# The Role of Impulsivity and Self-Control in Suicidal Ideation and Suicide Attempt

**DOI:** 10.3390/ijerph20065012

**Published:** 2023-03-12

**Authors:** Aleena Martin, Mitchell Oehlman, Jacinta Hawgood, John O’Gorman

**Affiliations:** Australian Institute for Suicide Research and Prevention, WHO Collaborating Centre for Research and Training in Suicide Prevention, School of Applied Psychology, Griffith University, Brisbane, QLD 4122, Australia

**Keywords:** suicidality, buffering hypothesis, risk factor, protective factor, perceived stress

## Abstract

Two studies are reported examining the relation of self-control, as measured by self-report inventories, to indices of suicidal ideation and suicide attempts. In the first study (*n* = 113), self-control related significantly (*p* < 0.05) and negatively to both indices (*r* = −0.37 and *r* = −0.26), and, in a hierarchical regression analysis, added significantly to the variance in the suicidal ideation index accounted for by a measure of impulsivity. The second study (*n* = 223) replicated the findings of the bivariate correlations (*r* = −0.55 and *r* = −0.59) with the suicidality indices in the first study, both with the earlier measures and with alternative measures of self-control and impulsivity. Results indicated self-control added to the prediction of both indices and not just the ideation index. The second study also demonstrated that self-control acts as a moderator for perceived stress, a known risk factor for suicidality, such that, at low levels of perceived stress, there is little difference between those high and low in measured self-control, but that at high stress levels, those with high self-control had lower scores on suicidal ideation. The results are interpreted as showing that self-control is a protective factor for suicidality.

## 1. Introduction

Suicide, the intentional termination of one’s own life [1], is a serious public health issue with more than 700,000 people dying by suicide each year [2]. The impacts of suicide are far-reaching. For every life lost through suicide, an additional 135 people are directly or indirectly impacted, and may require clinical or support services [3]. Suicide is often preceded by suicidal ideation, fluctuating thoughts directed to an individual’s intentions to kill themself with or without a plan [1,4]. Suicide may also be preceded by one or more suicide attempts, non-fatal self-injurious behaviours enacted purposefully with or without the intention of a fatal outcome [1].

Prevention efforts have largely been directed to identifying risk and protective factors for suicide, suicidal behaviour, and suicide attempts. A risk factor is defined generally as a “characteristic at the biological, psychological, family, community, or cultural level that precedes and is associated with a higher likelihood of problem outcomes” [5] (p. xxviii). A protective factor is “a characteristic at the biological, psychological, family, or community level that is associated with a lower likelihood of problem outcomes or that reduces the negative impact of a risk factor on problem outcomes” [5] (p. xxvii). Risk factors for suicide, suicidal ideation, and suicide attempts have been more extensively studied than protective factors for these events, although the results to date from both studies of risk factors and studies of protective factors have been disappointing [6].

The work reported here was undertaken to investigate the possible role of self-control as a protective factor for suicidal ideation and suicide attempt, in both senses of the definition offered by O’Connell [5], that is, as a factor negatively related to indices of suicidality, and in the sense of reducing the impact of a risk factor. Recent theory and research have indicated the importance of distinguishing between ideation and attempt in terms of the factors affecting them. Ideation to action theories of suicidal behaviour, such as those of Joiner, Klonsky, and O’Conner [7], highlight differences between the onset of ideation and the likelihood of translating thoughts into actions. We sought to recognise this distinction in the assessment process in our studies by employing a measure focussed on suicidal ideation (the Suicide Ideation Attributes Scale, SIDAS) [8] and one predominantly directed to suicidal behaviour (the Suicidal Behaviours Questionnaire-Revised, SBQ-R) [9].

Self-control was selected as the target construct for two main reasons. First, there is extensive literature in contexts other than suicide research indicating the importance of self-control for life outcomes, including health, educational attainment, labour market performance, assets and savings, and life satisfaction [10,11,12]. People with higher levels of self-control are generally found to enjoy better health and well-being, and greater success in life. However, there have been few studies to date linking self-control with suicidal ideation or suicide attempts. Mathew and Nanoo [13] reported that the 7-item self-control scale from the Ways of Coping Questionnaire [14] differentiated adolescents who had attempted suicide from those who had not. Fergusson and colleagues [15], in a longitudinal study, followed 1265 New Zealand children aged 6 to 12 years until they were 30 years old. Childhood self-control was assessed with the Rutter Behaviour Questionnaire [16]. The study found that lower self-control was related to greater suicidal ideation. Moffitt et al. [17], in a longitudinal study, also measured self-control with a variety of instruments and found that children with poor self-control were more likely to attempt suicide in adulthood.

The second reason for targeting self-control as a possible protective factor was that self-control can be conceptualised as the opposite of impulsivity, a construct frequently implicated as a risk factor for suicide and suicidal behaviour, e.g., [18,19]. Early analyses of the Barratt Impulsivity Scale (BIS-II) [20], the most widely used measure of impulsivity in suicide research, indicated a subfactor of the scale that researchers labelled Self-control [21]. Subsequent analyses have focused on a three-subfactor solution to the item structure of the scale, and the contribution of self-control has been overlooked. The question arises whether impulsivity is important because low scores on the BIS-II indicate greater self-control and it is this that is responsible for any relations with suicidal ideation of suicide attempts.

Mamayek and colleagues [22] argued that impulsivity and poor self-control are not synonymous, but rather two distinguishable concepts. Acting impulsively by giving in to desire, they considered, involved a different process to inhibiting the desire. Hofmann et al. [23] proposed that impulsivity and self-control reflect the action of different cognitive systems in a variant of the dual-systems approach to understanding behaviour. The impulsive system is the faster acting system, with the self-control system relying on reflective, language driven processes. This line of thinking implies that impulsivity and self-control make separate contributions to suicidal ideation and behaviour.

Two studies are reported here. The first sought to establish whether a measure of self-control is related to measures of suicidal ideation and suicide attempt, and whether this relation reflects more than the simple overlap of self-control with any common variance between impulsivity and the suicide indices. The Multidimensional Self-Control Scale (MSCS) [24] was used to assess self-control. The MSCS is a psychometrically sound scale based on the work of de Ridder et al. [25] who demonstrated a distinction between self-control that improved engagement in desired or goal-directed behaviour (initiatory self-control) and self-control involved in refraining from undesired behaviour (inhibitory self-control). The two scales were correlated in the de Ridder et al. study (0.68 and 0.66 in two samples), but initiatory self-control was a better predictor of positive behaviours, such as hours spent studying or exercising, whereas inhibitory self-control was a better predictor of regulating undesired behaviours, such as smoking or alcohol use. 

Measures of impulsivity (BIS-II), self-control (MSCS), suicidal ideation (SIDAS), and suicide attempts (SBQ-R) were administered in a cross-sectional study of adult community volunteers. It was hypothesised that: (a) based on previous findings, impulsivity is related positively and self-control negatively to suicidal ideation and attempts; and (b) based on the argument that different systems are involved in impulsivity and self-control, self-control adds to the prediction of suicidal ideation and attempts over and above the contribution of impulsivity.

The second study was designed as (a) a replication of the findings of Study 1, and (b) an extension of the findings to a possible role for self-control as a protective factor in buffering the effects of stress on suicidal thinking and behaviour. We used the self-control and impulsivity measures employed in Study 1 to examine the replicability of the earlier results, but sought to extend the findings by using the Brief Self Control Scale (BSCS) as well [26], widely used in the literature of social psychology, and the SUPPS-P Impulsive Behaviour Scale [27]. This is a shorter version of the UPPS-P [28] that assesses impulsivity across five sub-domains: negative urgency, positive urgency, and sensation seeking. The extension of findings was based on the buffering hypothesis advanced by Johnson et al. [29].

Johnson et al. [29], on the basis of a meta-meta-analysis of protective factors for suicidality, proposed that protective factors are distinct from risk factors rather than simply representing low levels of risk. The hypothesis proposes that protective factors moderate (i.e., interact with) suicide risk in such a way that the impact of risk is reduced. In the case of an individual who is not at risk of suicide, the protective factor is irrelevant. However, in the presence of risk, the previously dormant or irrelevant protective factor reduces or buffers the effect of risk, weakening or eliminating its relationship with suicidality.

Several risk factors have been employed in tests of this buffering hypothesis, but most studies have used factors strongly implicated in the onset of suicidality, including hopelessness, depression, and stress. For example, an influential study by Johnson and colleagues [28] tested positive schematic appraisals as a moderator of the positive relation between stressful life events and suicidal behaviour. The study found that individuals who scored highly on a measure of positive schematic appraisals were less likely to experience suicidal behaviours (ideation or attempt) than those with low scores in positive schematic appraisals when stress was high, but not when stress was low.

The risk factor chosen for Study 2 was perceived stress as measured by the Perceived Stress Scale (PSS) [30] that assesses perceived stress over the past month, lack of perseverance, and lack of premeditation. 

We hypothesised that the relations between impulsivity and self-control and the suicidality indices that were observed in Study 1 would be found again, with the measures used previously and with the new measures. A second hypothesis was that self-control acts as a moderator between stress and suicidality such that only at high levels of stress is there a strong relation with self-control.

## 2. Method

### 2.1. Participants

A total of 149 participants agreed to participate, but 12 participants failed to provide any data, and a further 24 (16%) provided demographic data, but did not respond to any item on one or more of the questionnaires. These 36 participants were excluded from the analysis, leaving 113 as the final sample for analysis. The sample included 29 males and 83 females (one participant did not complete the gender question), and of these, 99 reported being employed, 14 reported being unemployed, 57 reported being in a relationship, and 56 reported being single. The average age of participants was 41.3 years (*SD* = 15.72). The differences on the four demographic variables between the 24 who did not complete and the 113 who did were small (e.g., 3.8 years in age and 5% in the frequency of females) and in no instance statistically significant.

To replicate the findings of Study 1 that MSCS adds a statistically significant 7% to variance predicted by BIS-II to provide a final R^2^ of 0.158, power analysis using G-Power [31] indicated that a sample size of 116 was required (α = 0.05, power = 80%). With an attrition rate of 30%, the total N required was estimated at 169. This was sufficient, according to G-Power, to find the smallest of the correlations (r = 0.24) between self-control, impulsivity, and the suicidality indices statistically significant. A total of 284 agreed to participate in the study, but 13 did not begin, and 48 did not complete most of the questions, which left 223. Of these, there were some missing data for 26 participants. The sample included 38 males and 185 females, and of these, 178 reported being employed part-time or full-time, 45 reported being unemployed, 119 reported being in a relationship, and 104 reported being single. Average age was 42.4 years (SD = 13.56).

### 2.2. Procedure

Participants were recruited through online advertising by the Australian Institute for Suicide Research and Prevention on social media platforms (LinkedIn, Twitter, and Facebook). The 30 min online self-report survey was built in the Research Electronic Data Capture program [32] and provided to participants via a link. Consent was indicated by beginning the survey after reading the information page. All adults were invited to participate with no restrictions other than the requirement to be 18 years of age. As an incentive, participants had a chance to win either an AUD100 or AUD50 Mastercard gift card. An external link was provided for prize entry to ensure anonymity of survey responses. The procedure followed was essentially that in Study 1.

### 2.3. Measures

Suicidal ideation was measured using the SIDAS, a 5-item self-report questionnaire about suicidal ideation in the past month. The severity of past suicidal behaviours and attempts was measured using the SBQ-R, a 4-item self-report scale.. The MSCS is a 29-item scale. The BIS-11 is a 30-item self-report scale scored for three components of impulsivity and their total. 

For Study 2, three scales were added to those used in Study 1: the BSCS (a 13-item scale), the SUPPS-P (a 20-item scale), and the PSS (a 10-item scale). 

## 3. Results

### 3.1. Study 1

#### 3.1.1. Missing Data Analysis

For the 113 participants, there were missing data on 59 of the 68 questionnaire items (87%), and for 19 of the 113 cases (17%). Little’s Missing Completely at Random (MCAR) test [33] was consistent with the data missing completely at random, χ^2^ = 9.21, *p* > 0.05. The expectation maximisation (EM) procedure in SPSS was used to replace the missing data for the 113 cases. The analyses were also run using the option to delete missing data listwise. The findings of statistically significant and non-significant effects were the same for the two types of analysis, except that the analysis of SIDAS with component scores did not show a significant incremental effect, *F*(2, 107) = 4.326, *p* = 0.016. The EM results are reported here.

#### 3.1.2. Descriptive Statistics

All variables for analysis were checked for the assumptions of regression. All were within tolerable limits except for SIDAS, which had a strong positive skew (z = 7.42, *p* < 0.001). We tried a number of transformations with little effect on the skew. A histogram of the standardised residuals and a plot of the standardised residuals against the standardised predicted values both showed departures from normality. The Kolmogorov–Smirnov and Shapiro–Wilk tests of a normal distribution of residuals both rejected the null hypothesis (KS = 0.187, *df* = 113, *p* < 0.001; SW = 0.863, *df* = 113, *p* < 0.001) consistent with a non-normal distribution. We have added this information for the reader, and have added a comment on the need for caution given this failure of an assumption of regression. The results are reported for the raw SIDAS scores.

Internal consistency of the measures in the present study was estimated using Cronbach’s coefficient alpha. For SIDAS, alpha = 0.8; for SBQ-R, alpha = 0.79; for the MSCS, alpha = 0.91; for inhibitory self-control, alpha = 0.93; for initiatory self-control, alpha = 0.84; for the BIS-11, alpha = 0.84 for the full scale, alpha = 0.78 for attentional impulsivity, alpha = 0.63 for motor impulsivity, and for non-planning impulsivity, alpha = 0.67.

Table 1 summarises the means, standard deviations, and intercorrelations for all variables. 

Inspection of Table 1 indicates that the suicidality indices are strongly related. Impulsivity and self-control measures relate to both suicidality indices, impulsivity positively and self-control negatively, as predicted. The Attentional Impulsivity component was the component of the three impulsivity components that related at a statistically significant level to the two suicidality indices. The Inhibitory Self-Control component and not the Initiatory component was the self-control component that was the significant correlate of the suicidality indices. Attentional Impulsivity related strongly to Inhibitory Self-control, almost as strongly as it did to total score on BIS-II. That is, the two measures share considerable common variance.

#### 3.1.3. Hierarchical Regression Analyses

To assess if self-control adds to the prediction of the suicide indices from impulsivity alone, a hierarchical multiple regression was applied, with BIS-II entered at the first step, and MSCS at the second. With SIDAS as the outcome variable, *R*^2^ at the first step was 0.084, *F*(1, 111) = 10.12, *p* = 0.002, and at the second step was 0.158, *F*(1, 110) = 9.66, *p* = 0.002. The increment in variance 7% was statistically significant, *F*(1, 110) = 9.659, *p* = 0.002. With SBQ-R as the outcome variable, *R*^2^ at the first step was 0.054, *F*(1, 111) = 6.357, *p* = 0.013, and at the second step was 0.075, *F*(1, 110) = 2.44, *p* = 0.121. The increment in variance accounted for (2%) was not statistically significant, *F*(1, 110) = 1.224, *p* = 0.121.

The analyses were repeated using the components of BIS-II and MSCS as predictors in place of the total scores. With SIDAS as the outcome variable, *R*^2^ at the first step (with the BIS-II components) was 0.174, *F*(3, 109) = 7.66, *p* < 0.001, and at the second step (with the addition of the MSCS components) was 0.236, *F*(2, 107) = 4.33, *p* = 0.016 (i.e., an increase of 6% in the variance accounted for). With SBQ-R as the outcome variable, *R*^2^ at the first step was 0.1, *F*(3, 109) = 4.04, *p* = 0.009, and at the second step was 0.120, *F*(2, 107) = 1.21, *p* = 0.302 (i.e., an increase of 2% in the variance accounted for).

A summary of the analysis in the case of SIDAS is presented in Table 2. The Attentional component of the BIS-II was the only statistically significant component at Step 1 of the analysis but, at Step 2, the Inhibitory component of the MSCS became the only statistically significant component.

### 3.2. Study 2

#### 3.2.1. Missing Data Analysis

For the 223 participants, there were missing data on all the variables (but in no case was this as much as 10%) and for 26 of the 223 cases (12%). Little’s MCAR [33] was consistent with the data missing completely at random, χ^2^ = 2650.46, *p* = 0.623. The expectation maximisation (EM) procedure in SPSS was used to replace the missing data. The analyses were also run using the option to delete missing data listwise. The findings of statistically significant and non-significant effects were the same for the two types of analysis. The EM results are reported here.

#### 3.2.2. Descriptive Statistics

A check of variable distributions showed SIDAS to have strong positive skew (z = 6.20, *p* < 0.001).

Internal consistency estimates for the new measures used with this sample were for the BSCS, alpha = 0.86; for the SUPPS-P, alpha = 0.87; and for the PSS, alpha = 0.81.

Table 3 presents means, standard deviations, and intercorrelations for total scores on the measures used in this study, and Table 4 presents these statistics for the component scores of the impulsivity and self-control measures. Inspection of Table 3 indicates that both suicidality indices are strongly related. Impulsivity as measured by both the BIS-II and the SUPPS-P relates positively to SIDAS and to SBQ-R, and that self-control as measured both by MSCS and BSCS relates negatively to both indices.

Inspection of Table 4 indicates that Attentional Impulsivity is the component of the BIS-II with the strongest correlation with SIDAS and SBQ-R, as in Study 1, but unlike the results of Study 1, it is not the only component of BIS-II to relate to the suicidality indices. It is also the case that, unlike the results of Study 1, both components of MSCS relate at a statistically significant level to SIDAS and SBQ-R. Attentional Impulsivity again correlated strongly with Inhibitory Self-control and at a level only slightly below the level of correlation between that for each component and the total score of which it is a part, pointing to little discrimination for components of ostensibly different measures.

For the measures added to this study, correlations of the SUPPS-P components with SIDAS and SBQ-R were statistically significant for all components, except for Sensation Seeking, where the correlations for both indices were close to zero. Of the remaining components, Negative Urgency showed the strongest correlations. It correlated with both Attentional Impulsivity and Inhibitory Self-control.

#### 3.2.3. Hierarchical Regression Analyses

A series of hierarchical regression analyses were run to examine the additional contribution of the self-control measures to the impulsivity measures in predicting both SIDAS and SBQ-R. The first was an attempted replication of the finding in Study 1 that MSCS adds to the prediction from BIS-II. With BIS-II entered at the first step, and SBQ-R as the outcome variable, *R*^2^ was 0.22, *F*(1, 221) = 67.71, *p* < 0.001, and at the second step was 0.35. The increment in variance accounted for (13%) was statistically significant, *F*(1, 220) = 45.03, *p* < 0.001. With SIDAS as the outcome variable, *R*^2^ at the first step was 0.17, *F*(1, 221) = 44.04, *p* < 0.001, and at the second step was 0.30. The increment in variance accounted for (13%) was statistically significant, *F*(1, 220) = 40.93, *p* < 0.001.

With the components of the two variables as predictors and for SBQ-R as the outcome variable, *R*^2^ for Attentional Impulsivity, Motor Impulsivity, and Non-Planning at the first step was 0.24, *F*(3, 219) = 23.56, *p* < 0.001. At the second step, with Inhibitory and Initiatory Self-Control added to the equation, *R*^2^ was 0.38, a statistically significant change, *F*(2, 217) = 23.56, *p* < 0.001. For SIDAS as the outcome variable, *R*^2^ for the three impulsivity components was 0.18, *F*(3, 219) = 15.85, *p* < 0.001, and with the two self-control variables added was 0.31, also a statistically significant change, *F*(2, 217) = 20.46, *p* < 0.001. In both analyses, Attentional Impulsivity was the only component to make a statistically significant contribution at the first step. At the second step, the only statistically significant contributions (*p* < 0.001) were Inhibitory and Initiatory Self-Control. These results largely replicated those of Study 1 except that self-control contributed statistically significantly to both SIDAS and SBQ-R rather than to SIDAS alone.

In the second set of hierarchical regression analyses, the new impulsivity and self-control variables replaced those used in the first set. With SBQ-R as the outcome variable and SUPPS-P entered at the first step, *R*^2^ = 0.25, *F*(1, 221) = 71.79, *p* < 0.001. With BSCS entered at the second step, R^2^ = 0.29, a statistically significant change in the variance accounted for, *F*(1, 220) = 12.23, *p* < 0.001. With SIDAS as the outcome variable, and SUPPS-P entered at the first step, *R*^2^ = 0.15, *F*(1, 221) = 40.34, *p* < 0.001. With BSCS entered at the second step, *R*^2^ = 0.19, a statistically significant change in the variance accounted for, *F*(1, 220) = 9.44, *p* = 0.002. Thus, these results provide some evidence that self-control adds to the prediction of the suicidality measures over and above the contribution of impulsivity, and that this is the case with alternative measures to those used previously. The effect is thus not bound to particular tests of the constructs involved. The conclusion is qualified in that at the component level the effect is less clear. Negative Urgency, for example, was the only component making a statistically significant contribution (*p* < 0.001) to both SBQ-R and SIDAS, which mirrors the result for Attentional Impulsivity with which it is strongly correlated.

The third set of hierarchical regression analyses examined the role of self-control in moderating the relation between stress and the suicidality indices. A product term was formed by multiplying MSCS scores by PSS scores, which carried the interaction of self-control and stress. A summary of relevant descriptive statistics is presented in Table 5. MSCS and PSS were entered at the first step of a hierarchical regression analysis and the product term was added at the second step to assess whether including the interaction term increased the variance accounted for. With SIDAS as the outcome variable. *R*^2^ = 0.40, *F*(2, 220) = 73.53, *p* < 0.001, at the first step and *R*^2^ = 0.443, *F*(1, 219) = 16.78, *p* < 0.001, at the second step, which was statistically significant. With SBQ-R as the outcome variable, *R*^2^ = 0.437, *F*(2, 220) = 85.55, *p* < 0.001 at the first step and *R*^2^ = 0.438, *F*(1, 219) = 0.06, *p* = 0.811, at the second, which was not a statistically significant addition. 

With BSCS used to form the interaction term with PSS scores and SIDAS as the outcome variable, *R*^2^ at the first step was *R*^2^ = 0.374, *F*(2,220) = 65.25, *p* < 0.001, and at the second step was *R*^2^ = 0.40, *F*(1.219) = 10.104, *p* = 0.002, which was statistically significant. With SBQ-R as the outcome variable, at the first step was *R*^2^ = 0.410, *F*(2, 220) = 96.53, *p* < 0.001, which was statistically significant, and at the second step was *R*^2^ = 0.411, *F*(1, 219) = 0.371, *p* = 0.543, which was not statistically significant.

Figure 1 presents a plot of the significant interactions, with MSCS (left panel) and BSCS (right panel), and the stress measure. The low point in both panels is for PSS score 1 standard deviation below the mean, and the high point is for PSS score 1 standard deviation above the mean. Inspection indicates little difference between high and low self-control in SIDAS score at the low level of stress, but at the high stress point SIDAS score is lower for high self-control than for low.

## 4. Discussion

The studies reported here were predicated on the hypothesis that self-control acts as a protective factor for suicidality, given its positive role in health and well-being outcomes, and some previous findings on suicidal ideation and behaviour consistent with this. In the first study, the definition of a protective factor that was used was one that is linearly related to an outcome variable. Self-control as measured by the MSCS scale was found to meet the definition, with both the indices of SIDAS (suicidal ideation) and SBQ-R (predominantly suicidal behaviour) as outcome variables. It was further found that the MSCS scale added to the prediction of the outcome over and above that provided by the BIS-II, a widely used measure of impulsivity, at least in the case of SIDAS. The significance of this is that self-control is seen by some as the polar opposite of impulsivity, and its relation with suicidality would not, therefore, be surprising given the reported relation of impulsivity and suicidality. Instead, the present results point to a partly separate role for self-control, possibly more active and deliberate, in relation to suicidal ideation and behaviour. It may be that self-control strategies can be effective in the case of ideation, where deliberative control of thoughts is involved, but not in the case of behaviour, where stronger affective processes are engaged.

Self-control and impulsivity were correlated in the present study, and the Attentional and Inhibitory components of the two measures strongly so in Study 1, but self-control accounted for additional variance in SIDAS, implying that the two measures are not totally redundant at the level of total score. The attempted replication of this result in Study 2 was partly successful: self-control added to the prediction of both suicidality indices over and above the contribution of impulsivity, and this was true for both the measures used in Study 1 and the alternative measures introduced in Study 2. What did not replicate was the difference in finding when SBQ-R and not SIDAS was the outcome variable. In Study 1, the contribution of self-control was not statistically significant in the case of SBQ-R. In Study 2, the contribution of self-control was shown with both outcome variables. The simplest explanation for this is the difference in power of the two studies. Study 2 was a higher-powered study (larger N) and was thus more likely to reveal an effect for self-control as statistically significant.

At the component level, the pattern of findings varied between the two studies. Attentional Control appeared as the active component of impulsivity in both studies, but the stronger role for Inhibitory than Initiatory control found in Study 1 did not hold up in Study 2. The idea that the critical factor is effortful control of impulse, in terms of a two-process theory of self-control [22,23], was thus not clearly supported by the results of the second study.

The second study also examined the role of self-control as a protective factor in the sense of acting as a buffer against the risk for suicidality posed by stressful circumstances at least as perceived by the individual. The interaction effect between a protective factor and a stressor implied by the buffering hypothesis proposed by Johnson et al. [29] was demonstrated in Study 2 with both measures of self-control (MSCS and BSCS), but only with SIDAS as the outcome variable. This difference may have resulted from a difference in power in testing for the effect with the two measures. A larger N study is required to rule out this possibility.

The present studies have three limitations. One has to do with the nature of their design. Both were cross-sectional, and as such, any conclusions from them cannot sustain an inference about direction of causation, that is, that low self-control is responsible for lower values on the suicidality indices. This requires a longitudinal design, which would provide stronger grounds for such an inference.

The second limitation is that all measures were self-report, and thus share common method variance, and this, rather than actual relations among the behaviours implied by the measures, may be responsible for the relations, at least in part. Although self-report measures are easily obtained and can cover a number of behaviours, they are not the only way of gathering data relevant to the present hypothesis. Moffit et al. [17], for example, used a number of ways of assessing self-control in their study, including direct observation of behaviour, and some of these may provide an alternative to self-report. On the outcome side, momentary experience sampling is being used increasingly to map suicidal ideation and behaviour and, although it involves self-report, the reports are less prone to flaws in memory and possibly bias in reporting.

A third limitation was the non-normal distribution of scores on the SIDAS variable and the non-normal distribution of regression residuals, which is a violation of assumptions of regression. This indicates caution in interpretation of the findings, and the need in future research to employ an ideation measure that gives rise to a less skewed distribution.

The results of the present studies warrant further work replicating the finding of a buffering effect for self-control on suicidal ideation and possibly behaviour that might examine risk assessed by factors other than perceived stress, including hopelessness and depression. The literature on self-control includes the idea that self-control is a relatively stable characteristic of the person but does allow for self-control to develop with appropriate attention to the situational cues for particular behaviours. Self-control training, for example, has been found to be effective [34] and may be an option for populations in which there is elevated risk of suicidal ideation or behaviour.

One other point may be worth making. The implicit assumption in designing the studies and interpreting the outcomes reported here is that self-control is a positive influence, as shown by a variety of studies referenced earlier. However, it may be that over-control is itself a negative characteristic and one that poses a risk rather than providing a protection for suicidality. Perfectionism, for example, has been linked to an increased likelihood of suicidality [35], which implies compulsive control of thoughts and behaviour. Exploration of the broader network of control measures, including failure of control as with impulsive behaviour, adaptive control as reported by Tangney and colleagues [26], and obsessive or compulsive control as in perfectionism, may serve a better understanding of this important aspect of functioning and its relation to suicidality.

## 5. Conclusions

The results of the present studies point to a role for self-reported self-control as a protective factor for suicidal ideation and behaviour. High self-control was associated with lower scores on both the ideational and behavioural indices of suicidality and, for suicidal ideation, to be associated with lower scores when a risk factor for suicidality (perceived stress) was high. That is, self-control appeared to buffer the effect of the stressor. The effect for self-control was independent of its relation to impulsivity, with which it was correlated, and which was also related to suicidality. This outcome is consistent with the expression of suicidality being the result of both a fast-acting, impulsive system, and a slower, more deliberative system.

## Figures and Tables

**Figure 1 ijerph-20-05012-f001:**
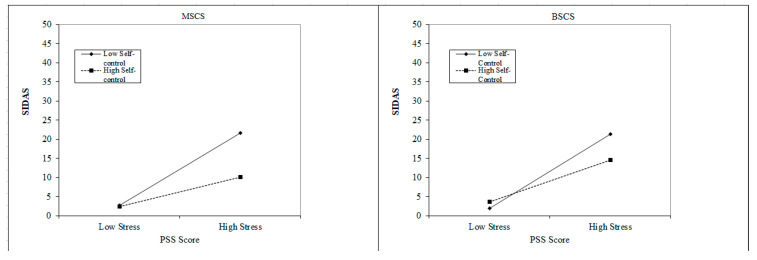
Interaction of Self-Control measured with the MSCS (**left panel**) and the BSCS (**right panel**).

**Table 1 ijerph-20-05012-t001:** Means, Standard Deviations, and Intercorrelations of Variables in Study 1.

Variable	*M*	*SD*	1	2	3	4	5	6	7	8
SIDAS	7.8	9.49								
SBQ-R	9.3	3.78	0.74							
Impulsivity (BIS-II)	63.8	9.11	0.30	0.24						
Attentional Impulsivity	17.6	3.79	0.41	0.28	0.79					
Motor Impulsivity	22.1	3.52	0.16	0.19	0.78	0.42				
Non-Planning Impulsivity	24.1	4.00	0.15	0.01	0.84	0.49	0.51			
Self-Control (MSCS)	95.9	14.91	−0.37	−0.26	−0.74	−0.70	−0.42	−0.66		
Inhibitory Self-Control	48.1	10.22	−0.44	−0.28	−0.74	−0.76	−0.43	−0.58	0.94	
Initiatory Self-Control	47.8	6.47	−0.17	−0.17	−0.55	−0.40	−0.29	−0.61	0.83	0.57

Note: *r* = > 0.185, *p* < 0.05, two-tailed.

**Table 2 ijerph-20-05012-t002:** Regression statistics at each step of the hierarchical analysis with the components of BIS-II and MSCS as predictors and SIDAS as the outcome variable.

		Step 1				Step 2		
Component	B	SE	*b*	*t*	B	SE	*b*	*t*
Attent Imp	1.124	0.257	0.446	4.378 *	0.522	0.325	0.207	1.606
Motor Imp	0.092	0.274	0.034	0.336	0.047	0.267	0.018	0.175
NonP Imp	−0.249	0.254	−0.105	−0.980	−0.552	0.283	−0.234	−1.954
Inhib SC					−0.352	0.132	−0.384	−2.659 *
Init SC					−0.071	0.161	−0.049	−0.445

Note: Attent Im*p* = Attentional Impulsivity (BIS-II), Motor Im*p* = Motor Impulsivity (BIS-II). NonP Im*p* = Non-Planning Impulsivity (BIS-II), Inhib SC = Inhibitory Self-Control (MSCS), Init SC = Initiatory Self-Control (MSCS). * *p* < 0.05.

**Table 3 ijerph-20-05012-t003:** Means, Standard Deviations, and Intercorrelations for whole scores used in Study 2.

Variable	*M*	*SD*	1	2	3	4	5
1. SIDAS	11.7	13.58					
2. SBQ-R	10.5	4.33	0.75				
3. Impulsivity (BIS-II)	66.9	12.80	0.41	0.47			
4. Self-control (MSCS)	89.4	19.69	−0.55	−0.59	−0.73		
5. Impulsivity (SUPPS-P)	44.3	10.19	0.39	0.50	−0.79	−0.70	
6. Self-control (BSCS)	39.1	9.31	0.41	−0.49	−0.74	0.76	−0.72

Note: All coefficients *p* < 0.05, two tailed.

**Table 4 ijerph-20-05012-t004:** Means, Standard Deviations, and Intercorrelations of the component scores in Study 2.

Variable	*M*	*SD*	1	2	3	4	5	6	7	8	9	10	11
1. SIDAS	11.7	13.58											
2. SBQ-R	10.5	4.33	0.75										
3. Attentional Impulsivity	19.1	4.72	0.39	0.47									
4. Motor Impulsivity	22.9	4.63	0.31	0.37	0.53								
5. Non-Planning	24.9	5.72	0.33	0.37	0.54	0.66							
6. Inhibitory Control	44.0	12.31	−0.51	−0.57	−0.74	−0.55	−0.65						
7. Initiatory Control	45.4	10.10	−0.44	−0.45	−0.38	−0.29	−0.54	0.54					
8. Negative Urgency	10.9	3.26	0.50	0.62	0.66	0.53	0.59	−0.76	−0.54				
9. Lack of Perseverance	7.6	2.34	0.17	0.25	0.29	0.32	0.50	−0.49	−0.49	0.35			
10. Lack of Premeditation	8.0	2.70	0.30	0.38	0.56	0.67	0.75	−0.64	−0.52	0.60	0.56		
11. Sensation Seeking	9.3	3.05	−0.01	0.01	0.01	0.32	0.15	−0.04	0.07	0.00	0.13	0.18	
12. Positive Urgency	8.5	3.16	0.38	0.44	0.52	0.56	0.55	−0.60	−0.27	0.59	0.32	0.61	0.31

Note: *r* ≥ 0.132, *p* < 0.05, two-tailed.

**Table 5 ijerph-20-05012-t005:** Means, Standard Deviations, and Intercorrelations for variables used in the Moderated Regression Analyses.

Variable	*M*	*SD*	1	2	3	4
SIDAS	11.7	13.58				
SBQR	10.5	4.33	0.748			
Stress	22.3	7.56	0.607	0.620		
MSCSxStress	1896.5	549.66	0.279	0.316	0.735	
BriefSCxStress	833.1	255.34	0.330	0.336	0.698	0.833

Note: All coefficients *p* < 0.05 two tailed.

## Data Availability

The data for this project were gathered with the explicit promise to participants that individual responses would not be published in any form, only aggregated data, because of the sensitivity of the information they were providing. This undertaking formed part of the conditions under which the project was approved by the institutional ethics committee.

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
