# Peer review of "The Role of Impulsivity and Self-Control in Suicidal Ideation and Suicide Attempt"

_ijerph, 2023, doi:10.3390/ijerph20065012_

Round 1

Reviewer 1 Report

Dear Authors, 

In general the paper is well written, presents a relevant topic and the theoretical framework and discussions presented are correct for the topic they develop. However, there is an aspect that generates some confusion to the reader, from my point of view, it would be necessary to unify a little more both studies, allowing a better understanding of the paper in general, or, to separate completely both, although to show some relationship between them, this would be the least advisable option. Therefore, I recommend unifying at least the methodology and discussion sections, even if the results are shown in two parts. I believe this will improve understanding. 

Best regards, 

The reviewer

Reviewer 2 Report

Thank you to the authors for this article. I offer some comments and suggestions that I hope help to improve the article.

With regards to p < .05, I offer an argument as to why the authors should not do this:

1. Principle 3 of the American Statistical Association’s statement on p values states that ‘Scientific conclusions and business or policy decisions should not be based only on whether a p-value passes a specific threshold.’ See: Wasserstein, R. L., & Lazar, N. A. (2016). The ASA statement on p-values: context, process, and purpose. The American Statistician70(2), 129-133.

2. Ronald Fisher, who introduced the concept of nominal (nominating p < 0.05 as a significance level), levels of significance said “No scientific worker has a fixed level of significance at which from year to year, and in all circumstances, he rejects hypotheses; he rather gives his mind to each particular case in the light of his evidence and his ideas.” See Fisher RA. Statistical methods and scientific inference. Edinburgh: Oliver and Boyd; 1956. – I apologise for the male pronouns in the quote I cannot change.

3. Neyman and Pearson, who invented the null hypothesis significance testing paradigm, said: “it is doubtful whether the knowledge that [a P value] was really 0.03 (or 0.06), rather than 0.05…would in fact ever modify our judgment” See Neyman J, Pearson ES. On the use and interpretation of certain test criteria for purposes of statistical inference: part I. Biometrika. 1928;20A:175–240

4. Fisher suggested p < 0.05 because it was convenient to calculate by hand (being close to 2 standard deviations away from the mean) because he didn’t have a computer. See Kennedy-Shaffer, L. (2019). Before p< 0.05 to beyond p< 0.05: using history to contextualize p-values and significance testing. The American Statistician, 73(sup1), 82-90.

5. Related to point 4, it’s an arbitrary threshold and a declaration of ‘statistical significance’ has been called meaningless by the authors of that American Statistical Association statement: See Wasserstein, Ronald L., Allen L. Schirm, and Nicole A. Lazar. "Moving to a world beyond “p< 0.05”." The American Statistician 73.sup1 (2019): 1-19.

6. Reliance on the null hypothesis significance testing paradigm has ignored patterns in data that led to the maintenance of policies that conferred increased morbidity and mortality, such as the right turn on red policy in traffic safety, and the withholding of risk-reducing interventions, such as shoulders on the road. See Hauer, E. (2004). The harm done by tests of significance. Accident Analysis & Prevention, 36(3), 495-500.

7. The P value is a test of how extreme the data are assuming that random sampling occurred and the null hypothesis is true. Why random sampling? Because Ronald Fisher’s applications were agricultural and industrial, and it’s very easy to draw repeated random samples in those applications. This paragraph explains:

“The preoccupation with significance testing derives from the research interests of the statisticians who pioneered the development of statistical theory in the early 20th century. Their research problems were primarily industrial and agricultural, and they typically involved randomized experiments or random-sample surveys that formed the basis for a choice between two or more alternative courses of action. Such studies were designed to produce results that would enable a decision to be made, and the statistical methods employed were intended to facilitate decision-making. The concepts that grew out of this heritage are today applied in clinical and epidemiologic research, and they strongly reflect this background of decision-making.” - Rothman, K. J., & Lash, T. L., (2021). Precision and study size. In Modern Epidemiology. Philadelphia: Wolters Kluwer.

These days, science is incremental and we don’t need to decide based on one study as to a course of action, which I know the authors know, given the mentions of replications later in the manuscript.

Anyway, the authors should not feel bad, as the literature is rife with statistical significance testing, even among statisticians. This history has also been erased from psychology textbooks on statistics: See Gigerenzer, G. (2004). Mindless statistics. The Journal of Socio-Economics33(5), 587-606.

The good news is, this is a simple fix. The authors can simply report the exact p-value, and simply think of it as a continuous measure as it is the same as weight or height or age. The p value is a statement about the data, and not about the null hypothesis, as it assumes the null hypothesis is true. So, you can say things like: the data are incompatible with the null hypothesis (p = 0.04) or the data were completely compatible with the null hypothesis (p = 1.00). What else can you do? Interpret the size of the regression coefficient, like how you would interpret Cohen’s D. I don’t know of guidance for this sorry, but you may find some. Again, Cohen’s effect size levels were suggestions and also arbitrary thresholds, but the effect size is more important than the p-value, as the p-value confounds study size and effect size (see Lang, J. M., Rothman, K. J., & Cann, C. I. (1998). That confounded P-value. Epidemiology (Cambridge, Mass.)9(1), 7-8.), and you want to report unconfounded measures where possible. What was the change in variance before and after the introduction of self-control? That’d be good to put in the abstract, along with the precise p-value for the change.

This is the first article I’ve peer-reviewed that has cited 700,000 deaths per year rather than 800,000, the old WHO estimates as the authors will know. That is great!

Page 1, line 28: Wouldn’t suicide always be preceded by suicidal ideation? Unless you’re defining ideation as some consideration for a defined period of time.

Page 1, lines 42 to 42: You may just like to clarify what you mean by ‘both types of studies have been disappointing’? In terms of predicting suicidal behaviour?

Page 2, line 45: Do you need to include a reference for O’Connell? I presume it is reference 5. Page 2, lines 50 to 54: The text itself is fine, but as it’s the measures you used, it should go in the methods in 2.1.3? This would avoid some repetition.

Page 3, line 116: ‘joined’ is a bit ambiguous. Perhaps ‘agreed to participate’? Can’t do much with those 12. Could you add (16%) after ‘24’? Just to flag with the authors for future studies that there is a framework for handling missing data in observational studies: Lee, K. J., Tilling, K. M., Cornish, R. P., Little, R. J., Bell, M. L., Goetghebeur, E., ... & Carpenter, J. R. (2021). Framework for the treatment and reporting of missing data in observational studies: The Treatment And Reporting of Missing data in Observational Studies framework. Journal of clinical epidemiology, 134, 79-88.

You could ask if the 24 people were different demographically from the 113 who participated.

Page 3, line 120: ‘complete’ might read better than ‘endorse’?

Page 3, line 122: Just add ‘The’ before ‘Average’

Page 3, 2.1.3 Measures: Unless the journal instructs otherwise, the internal consistency is data from this study and so should go in results?

Page 3, line 145: Just need to say ‘missing completely at random (MCAR)’ instead of ‘MCAR’, then can say ‘MCAR’ just below.

Page 3, line 146: Could you please report the exact p-value for the test?

Page 4, line 150: It’d be great to report the two p-values here and perhaps standard errors, to understand if the difference is a function of increased precision in the imputed dataset. The change may not be much. E.g., Gelman, A., & Stern, H. (2006). The difference between “significant” and “not significant” is not itself statistically significant. The American Statistician, 60(4), 328-331.

Page 4, line 147 to 151: regression assumes that the errors (residuals) are approximately normally distributed. There are also 5 other assumptions that you need to test and can do as part of the output of a regression analysis in SPSS. You can read about them in the Laerd Statistics guide on multiple regression in SPSS, which is free for academics to subscribe to but costs money for students (I have no affiliation with Laerd Statistics).

Page 4, line 173: Instead of this line here, you could just say ‘The increment in variance was 7%, F(1, 110) = 9.659, p = .002.’ If I told you that you were going to get a pay rise, would you want to know whether it was statistically significant, or how much more (7%) it was? See Valentine, J. C., Aloe, A. M., & Lau, T. S. (2015). Life after NHST: How to describe your data without “p-ing” everywhere. Basic and applied social psychology, 37(5), 260-273.

Page 4, line 175 to 176: Instead of this line here, I suggest ‘The increment in variance for (2%) was 5% smaller, but still incompatible with a null hypothesis of no difference between steps, F(1, 110) = 1.224, p = .121’ – note that a P value > 0.5 would suggest that the data were more compatible with the null than not.

Page 5, line 179: Sorry, but is it possible to report the change in variance as a percentage? I don’t do linear regression much, but I thought you could express incremental variance explained as a percentage. What would be more important to know is what the original variance explained was and what the revised variance explained is, rather than the p value, which I can already see.

Page 5, line 190 to 200: I would not worry about the multinomial logistic regression currently. The rationale:

-          You need to look at the errors (residuals). I recommend looking at Laerd Statistics for guidance on this. So, you can’t say that you’ve violated this assumption and need to rectify it yet.

-          The p value tests all the assumptions used to compute the model, one of which is random sampling, as explained earlier. Since most observational studies like this don’t draw random samples, it just means you need to interpret your results more cautiously as you will always not be meeting one assumption unless you randomly sample from the population you’re making inferences about.

-          Multinomial logistic regression has its own set of assumptions, which aren’t discussed.

-          I would have thought transforming the variable (e.g., sqrt or logarithm) would be a first-line approach to dealing with non-normality, rather than trichotomising it. But as discussed, we don’t know about your residuals so we can’t confirm this assumption is violated.

Page 5, line 206: I would not worry about using ‘predictive’ terminology in a cross-sectional study. You could say ‘explain the variance’ or ‘explain the association’.

Page 5, lines 208 to 213: The explanation here is plausible. But should you call the SBQ-R a measure of suicidality if it is about behaviour and not ideation? Could it also be that impulsivity has a much stronger association with behaviour than ideation and so sucks up all the variance? Acts are impulsive, but thoughts are perhaps intrusive rather than impulsive? Or, someone may think about something, but they are not impulsive unless they act on the thought? Impulsivity just seems more closely related to behaviour in my view.

Page 5, line 213: The replication is important yes, but if another study has a smaller sample but the same effect, their finding may not be statistically significant, and they might conclude that their findings are different from yours based on p > 0.05, when it is entirely about the sample size and imprecision, and not about the effect size (which might be the same). But the p-value confounds both so science might just go around and around in circles. It might be more cautious to call it ‘association’ and not ‘effect’.

Page 5, line 228: just a typo here ‘meta-metanalysis’.

Page 7, line 283: ‘consistent’ or ‘compatible’ are excellent terms to describe data in null hypothesis significance tests.

Page 8 – there is some more usage of ‘effect’ and ‘predict’ language on this page. Effects infer causal inference, and prediction does not, but both require longitudinal datasets in most cases and each have their own can of worms. For causal inference in cross-sectional studies, see Savitz, D. A., & Wellenius, G. A. (2022). Can cross-sectional studies contribute to causal inference? It Depends. American Journal of Epidemiology.

Page 9, line 369: It is really helpful to plot the interactions, thank you. But can the authors clarify if this is only plotting the significant interactions (selective plotting), or whether all interactions are plotted, which would be preferable.

Page 10, line 402: But the higher-powered study would have a better chance of detecting it wouldn’t it, not the lower-powered study? Related to type I and II errors, this is a great and short article to read about them: Rothman, K. J. (2010). Curbing type I and type II errors. European journal of epidemiology25, 223-224.

Overall, this article was well-written and easy and enjoyable to read.
